# On the Creation and Optical Microstructure Characterisation of Additively Manufactured Foam Structures (AMF) [note 1]

**DOI:** 10.3390/polym15173544

**Published:** 2023-08-25

**Authors:** Anselm Heuer, Maike Rees, Kay A. Weidenmann, Wilfried V. Liebig

**Affiliations:** 1Institute for Applied Materials—Materials Science and Engineering (IAM-WK), Karlsruhe Institute of Technology (KIT), Kaiserstrasse 12, 76131 Karlsruhe, Germany; uoxix@student.kit.edu (M.R.); wilfried.liebig@kit.edu (W.V.L.); 2Institute of Materials Resource Management (MRM), University of Augsburg, Am Technologiezentrum 8, 86159 Augsburg, Germany; kay.weidenmann@mrm.uni-augsburg.de

**Keywords:** porosity, entropy by Haralick, process-related pores, foam pores, fineness, degree of foaming, chemical blowing agent, Arburg plastic freeforming (APF)

## Abstract

Plastic-based additive manufacturing processes are becoming increasingly popular in the production of structural parts. Based on the idea of lightweight design and the aim of extending the functionality of additive structures, the production of additively manufactured foam structures has emerged as a new field of application. The optical characterisation of these structures is of particular importance for process adjustments and the identification of (unwanted) changes in the foam structure. The degree of foaming and the fineness of a foam structure are of interest at this point. In this context, only the part of a structure dominated by foam pores is considered a foam structure. So far, there are no sophisticated methods for such an optical characterisation. Therefore, in this work, microscope images of manufactured as well as artificially created additively manufactured foam structures were evaluated. On these images, the features porosity, pore size, pore amount and a measure for the textural change were determined in order to obtain information about changes within an additively manufactured foam structure. It is shown that additive structures show changing pore shapes depending on the orientation of the cutting plane, although there are no changes in the foaming behaviour. Therefore, caution is required when identifying changes within the foam structure. It was also found that, owing to the additive process, the total porosity is already set in the slicing process and remains constant even if the degree of foaming of individual tracks is changed. Therefore, the degree of foaming cannot be determined on the basis of the total porosity, but it can be assessed on the basis of the formation of large networks of process-related pores.

## 1. Introduction and Motivation

After plastic-based additive manufacturing overcame its origin in prototyping, structural and functional parts were also produced in the recent past. Usually, compared to other manufacturing processes such as injection moulding, plastic-based additive manufacturing processes have lower mechanical properties [1,2,3,4]. One reason for this is process-related pores, which is why previous research in this field focused intensively on the effects of these pores on the mechanical properties and how these pores can be reduced [4,5,6]. However, the deliberate creation of pores can also be useful. For example, porous structures often have better weight-related properties, and additive manufacturing can be used to produce lightweight structures quite cost effectively by only placing material where it is needed [7]. Another reason for deliberately adding pores is that functional parts with significant thermal or acoustic insulation can be created. Therefore, in addition to the additive manufacturing of pore-free parts, porous structures are also quite interesting.

A well-known process in plastic-based additive manufacturing is material extrusion (MEX), otherwise known as fused filament fabrication (FFF). Here, parts are built up layer by layer, and each layer in turn consists of individual extruded tracks. Usually, this process can create porous structures through gaps between the tracks, but this results in a coarsely distributed porosity, and the amount of welded joints between the tracks decreases. However, these welded joints have a large influence on the mechanical properties in addition to the general reduction in mechanical properties due to increased porosity [8,9]. Thus, increasing the porosity without reducing the amount of welded joints would be an attractive approach for producing porous structures for lightweight applications. This can be achieved by placing the pores inside the tracks so that the tracks are still in contact with each other. In plastics, the formation of such pores usually occurs through a foaming process in which gas bubbles form, grow and finally stabilise in the plastic [10]. Therefore, these pores are called foam pores in the following, while pores formed by gaps between the tracks and by the shape of the extruded tracks are called process-related pores.

Figure 1 illustrates this approach. Despite the fact that the tracks are all in contact, there exists significant porosity in the structure due to the foam pores within the tracks. In the following, such a structure is called an additively manufactured foam structure (AMF). In addition, it can be noted that for thin-walled structures in the order of the nozzle diameter, it is not possible to create significant porosity by gaps between the tracks.

So far, different approaches have been pursued to investigate the manufacturability of AMF. With the help of glass microballoons (GMBs) compounded into filaments, a porosity in an extruded track can be produced [11,12]. In this case, the porosity is not formed during manufacturing itself; it is already contained in the filament due to the GMB. The producibility and properties of such materials, known as syntactic foams, were investigated by Bonthu et al. [11,12]. Another approach is to use chemical blowing agents (CBAs) compounded into a filament. In this case, the decomposition of the CBA during the extrusion of a track produces gas and thus pores. For example, Damanpack et al. use this approach to foam polylactic acid in a MEX process [13]. Instead of CBAs, it is also possible to compound thermally expanded microspheres (TEMs) in a filament, which are expanded by an increase in temperature [14,15,16]. Kalia et al. show that a uniform foam structure can be achieved with TEMs in a MEX process [15]. In their review about AMF, Nofar et al. propose a filament-free approach by using the Arburg plastic freeforming (APF) process, which they find interesting because of the expected constancy of the foaming process [7]. According to their theory, there should be no loss of the dissolved blowing agent throughout the manufacturing process or during the melting of the plastic in the nozzle. Furthermore, plastic granules can be mixed directly with masterbatches containing CBA, eliminating the need to compound a filament with CBA or GMB.

Besides the investigation of the plain manufacturability of AMFs, the optical characterisation of the resulting foam structure is becoming increasingly important. From established foaming processes, such as foam injection moulding, it is known that the foam structure significantly affects the properties of a part [17]. Additional dependencies on the foam structure result from additive manufacturing: Nofar et al. show in their review of AMFs that the pore size is not constant over the cross section of an extruded track and depends on the process parameters of additive manufacturing [7]. Nofar et al. also show that the physical blowing agent concentration for a filament-based process decreases over the process time. This can significantly affect the foam structure during the process time. Deviating from the theory of Nofar et al., the foaming process in the APF process could still not be constant if masterbatches containing CBA are used. Since the machine performing the APF process has only one screw without mixing elements, the homogeneous mixing of plastic and masterbatch granules in the machine can be considered variable [18]. The use of a filament containing TMEs in a MEX process also shows a relationship between the process parameters and foam structure [14,16]. For example, the porosity of the foam structure increases as the nozzle temperature increases. Optical characterisation of the foam structure of AMFs is therefore the basis for further research into process–structure–property relationships. Since there have been established foaming processes, such as foam injection moulding, since the 1970s [19], the characterisation of foam structures is also not a new field of research. For example, in 1978, Villamizar et al. already measured the pore size in foam injection moulding and investigated the influence of the process parameters on the pore size in more detail [20]. Or in a recent review of Okolieocha et al. about microcellular and nanocellular foams, different measurable features are listed, including how they can be determined experimentally [21]. However, in established foaming processes, only the pores formed by the foaming process are usually present. In additive manufacturing, the characterisation of only the foam structure within the AMF is complicated by the simultaneously existing process-related pores (cf. Figure 1). For example, if a relationship between the CBA content and the degree of foaming or the fineness of the foam structure is to be determined (for an explanation of terms, see Section 2), the process-related pores should be ignored during the characterisation. Their size and shape can also be changed independently by the parameters of additive manufacturing [5]. A change in the process-related pores can also be incorrectly interpreted as a change in the foam structure. To make matters worse, the segmentation of the two pore types is partly not possible due to the visual similarity. This makes it necessary to adapt the established characterisation of foam structures such that it can be applied to AMFs.

Previous work on AMF focused on the manufacturability of AMF and carried out the initial qualitative optical characterisations of AMF. A determination of the typical features of foams (e.g., pore size or pore amount) and the interaction of the foam structure with the additively manufactured structure has not yet taken place. In this work, various image features are investigated for their usability in characterising AMF. The aim is to identify effects that can only be attributed to the additive manufacturing process and are not the result of the foaming process. Furthermore, interactions between the two structures are revealed. This will provide future research with methods to understand parameter influences on AMFs. In order to achieve this, microscope images of manufactured AMFs are evaluated. These AMFs are produced by the APF process by using a masterbatch containing a CBA. In addition, artificial images of AMF are created to enable the characterisation of difficult-to-manufacture structures. Instead of analysing individual selected microscope sub-images, the entire image is divided into many overlapping sub-images. Based on this, a quasi-continuous trend of a feature can be determined, which allows changes in the AMF to be exactly localised. Especially in additive manufacturing, this offers the potential to investigate structural changes layer by layer. First, important terms for understanding the work are explained, features from image processing are introduced, and the methods used are presented. Subsequently, the results of the image processing for the manufactured and artificial AMFs are presented and discussed with regard to the characterisation of AMFs.

## 2. Nomenclature for AMF

This section explains terms already used in established foaming processes as well as newly defined terms introduced for the characterisation of AMFs. This is to provide clarity on the meaning of the terms required for this work.



**Foam pores**



These are the pores that are formed through a foaming process in which gas bubbles form, grow and finally stabilise in the plastic. Owing to the way they are formed, they are located within extruded tracks and are usually spherical. In the literature on plastic-based additive manufacturing, such pores are also named intra-voids or intra-bead voids, whereby they do not necessarily have to be formed by a foaming process [5]. In Figure 2, foam pores are marked in red.



**Process-related pores**



These refer to the pores that are formed by gaps between the tracks and by the shape of the extruded tracks. In this work, the pore types, known in the literature on plastic-based additive manufacturing as raster gap voids, partial neck growth voids and infill voids [5], are summarised as process-related pores. Owing to the way they are formed, such pores can become very large and nonspherical. An additively manufactured structure always contains such pores in different sizes [22]. In Figure 2, process-related pores are marked in blue. 



**Foam structure**



Describes only the part of the entire structure that is influenced by foam pores. Accordingly, the foam structure describes only the inner part of the tracks. 



**Additively manufactured structure**



This describes a structure that has been manufactured by a plastic-based additive manufacturing process. Many parameters are discussed in the literature that influence the characteristic of this structure. For instance, the nozzle temperature, the amount of infill or the orientation of the tracks [5]. However, in order to distinguish the foam structure from the additively manufactured structure, the inner part of the extruded tracks is not considered a part of the additively manufactured structure. 



**Additively manufactured foam structure (AMF)**



This is the superposition of the foam structure and additively manufactured structure. 



**Degree of foaming**



This specifies the amount by which a quantity of plastic foams [19]. With the foamed volume $V_{\mathrm{Foam}}$, the degree of foaming is calculated according to
(1)$$\begin{array}{c} \mathrm{Degree}\;\mathrm{of}\;\mathrm{foaming}=\frac{V_{\mathrm{Foam}}-V_{0}}{V_{0}} \end{array}$$
and specifies the increase in percentage of the initial volume $V_{0}$. In addition to the degree of foaming, the term volume expansion ratio also exists in the literature [21]. It is defined as
(2)$$\begin{array}{c} \mathrm{Volume}\;\mathrm{expansion}\;\mathrm{ratio}\;\left(\mathrm{VER}\right)=\frac{V_{\mathrm{Foam}}}{V_{0}}. \end{array}$$

Okolieocha et al. use the VER to classify foams according to their density: high density (if VER≤4), medium density (if VER≥4–10), and low density (if VER≥10–50) [21]. For a constant mass, the degree of foaming and VER can also be calculated with the density. If the foamed and non-foamed structure in Figure 2 are compared, both quantities can be calculated for individual tracks. 



**Fineness**



This describes the distribution of pores for a given amount of porosity. Usually, the fineness is quantified with the pore density (number of pores in a given volume). If there is no information about the distribution in the volume, the pore density can also be estimated from the number of pores in a plane. Fineness is used to classify foams according to their foam structure: conventional foams 10^2^ Pores/cm${}^{3}$ to 10^6^ Pores/cm${}^{3}$, microcellular foams 10^9^ Pores/cm${}^{3}$ to 10^12^ Pores/cm${}^{3}$ and nanocellular foams greater than 10^15^ Pores/cm${}^{3}$ [21,23]. A theory is that the finer the foam structure, the higher the weight-related properties, as the influence of the pores becomes more and more negligible compared to pre-existing flaws in the plastic [23]. 



**Porosity**



The porosity ϕ indicates the percentage of an empty volume (usually filled with air) in a given volume. For foams, it can be calculated according to
(3)$$\begin{array}{c} \phi =\frac{V_{\mathrm{Foam}}-V_{0}}{V_{\mathrm{Foam}}}. \end{array}$$

In this work, porosity is distinguished according to the pore type (foam pores and process-related pores). The following applies as long as pores are clearly labelled:(4)$$\begin{array}{c} \phi_{\mathrm{Total}}=\phi_{\mathrm{Foam}}+\phi_{\mathrm{Process}-\mathrm{related}}. \end{array}$$

## 3. Features for the Characterisation of Foam Structures

The foam structure of a solid foam can be analysed on microscope images using image-processing methods. On such images, features are usually determined. In image processing, the term feature is used for a piece of information about the content of an image. If only averaged values of features are to be determined and no changes within the foam structure are to be examined, then the entire microscope image can be evaluated as a single image. Otherwise, an image of the entire structure is divided into different sub-images so that each sub-image can be analysed separately. These sub-images can either be qualitatively compared by a user, or image features are extracted and compared image by image. A quantitative analysis is preferable in order to be independent of the user’s experience. A porosity can be determined for foams, and in this case, the porosity is a feature of a sub-image [10]. Referring to the image of an AMF in Figure 1 resulting from the cutting plane, the total porosity is the ratio of the black pixels to the sum of all pixels within the part. In this case, information about the porosity exists only within the cutting plane (2D). If a segmentation of the pores into foam pores and process-related pores is possible, then the porosity can be specified separately for both pore types. Another feature of foams is the pore size. Pore size is often considered to be the diameter of a pore [7,24]. However, a definition of the diameter is not clear for irregularly shaped or coagulated pores. Since the process-related pores are not spherical but mostly irregularly shaped [25], in this work, the cross-sectional area of a pore is used as the pore size. In addition to the pore size, the pore amount is another feature. Related to the area of the cutting plane (2D), it can be used to characterise the fineness of a foam structure [7,21]. Besides these three basic features, other features, such as the amount of neighbouring pores to a pore, can be determined or whether the foam is primarily an open- or closed-cell structure [10]. However, the effort required for the determination of such features is increasing.

Instead of extracting typical features of foams, a sub-image can also be considered a texture, and texture features can be extracted from it using textural analysis methods. In the image processing of greyscale images, a texture can be considered the spatial variation of the grey value of the pixels [26]. There exist different texture analysis methods, which are more or less suitable depending on the application or image quality [27]. Sharma et al. conclude that the determination of the grey level co-occurrence matrix (GLCM) and the following extraction of features from this matrix are suitable to separate different patterns of a texture [28]. This GLCM encodes grey level differences between neighbouring pixels and was introduced by Haralick et al. together with the Haralick features [29]. These features are well-known tools in the classification of different images and have also been used to distinguish between different structures. For example, a classification of different manufactured food foams was investigated by Germain et al. [30]. They show that the pore size and pore size distribution are not sufficient to distinguish between different manufactured foams. However, since the configuration of the pores can be considered a texture, this is possible with the help of an analysis of the GLCM. Although it is not possible to determine what has changed based on such texture features, it is possible to determine exactly that something has changed.

Since neither the three presented features for foams nor texture features have been used for a quantitative characterisation of AMFs so far, the total porosity, pore size, pore amount and the entropy by Haralick as an example of a texture feature are used in the following.

## 4. Material and Methods

### 4.1. Material and Machine Used

AMFs were produced with the freeformer 200-3X from ARBURG GmbH + CoKG (Lossburg, Germany). This machine has a screw for feeding and plasticising plastic granules and, as a unique feature, a closable nozzle that can be opened and closed with a variable frequency. The machine operates according to the APF process, in which plastic in droplet form leaves the nozzle in an actively controlled manner, and the individual droplets weld together to form a droplet chain. In analogy to other plastic-based additive manufacturing processes, the droplet chain is placed on a movable build plate. The extruded track within a layer shown in Figure 1 corresponds to the droplet chain in the APF process. After completing a layer, the build plate is moved along the Z-axis, and the next layer is manufactured. This build plate is located inside a heatable chamber. Various nozzle diameters are available, and for this work, 0.2 mm was used.

The plastic used to manufacture the AMFs was acrylonitrile–butadiene–styrene (ABS) Terluran GP-35 granules from INEOS Styrolution [31]. The granules were dried at 80 ${}^{{}^{\circ}}C$ for 2.5 h before use. For the foaming behaviour, the CBA-containing masterbatch Clariant Hydrocerol ITP 815 from Polymer-Service PSG GmbH was mixed with the ABS granules. According to the data sheet, a masterbatch content of 0.8% to 2.5% is recommended for foaming [32]. A preliminary investigation by the authors to define a process field showed that higher concentrations are required in the APF process, which is why contents of 4% and 6% by weight were chosen, respectively. The masterbatch granules were not dried, as they do not absorb water. ABS and masterbatch granules have almost the same grain size and density such that segregation in the granulate hopper can be excluded.

### 4.2. Specimen Manufacturing of AMF and Image Preparation

A cuboid with a base area of 25 × 25 mm and a height of 57 mm was selected as the specimen shape. To investigate two different AMFs, two cuboids with a masterbatch content of 4% and 6% by weight were produced, respectively. The cuboids were sliced in the ARBURG freeformer software v2.30. A layer thickness of 0.26 mm, a droplet aspect ratio of 1.25 and a discharge rate of 100% was used. The droplet aspect ratio specifies the distance between the droplet chains within the layer (infill degree corresponds to 100%), and the discharge rate specifies the volume flow [22]. The amount of perimeters was set to one. An overlap between the infill and perimeter of 50% and a infill density of 100% was chosen. The infill was set to rectilinear with an orientation of ±45°. The printing speed for the infill was 65 mm/s, and for the perimeter, 20 mm/s.

The nozzle temperature was set to 240 ${}^{{}^{\circ}}C$, temperature at zone 2 to 200 ${}^{{}^{\circ}}C$, temperature at zone 1 to 170 ${}^{{}^{\circ}}C$ and the chamber temperature to 100 ${}^{{}^{\circ}}C$. The two zones refer to segments of the plasticating cylinder. Zone 1 is after the granulate hopper and zone 2 is around the dosing volume. The stagnation pressure was 50 bar, and the circumferential speed was 4 mm/min. In order to enable the optimal closing of the non-return valve on the screw, the decompression was not switched off and was left at the standard setting of 2 mm distance at a speed of 5 mm/min.

After the cuboids were manufactured, they were cut lengthwise, and an inner surface was ground and polished. The position and orientation of the cutting plane are shown in Figure 3a. Using the Axiovert 200 MAT microscope from Carl Zeiss AG, individual images were taken at an even distance above the specimen height, and these images were stitched into one image in Fiji [33] by using the Image Stitching [34] plugin. A section of this stitched microscope image is shown in Figure 3b. The resolution of the stitched images is 130 pixel/mm.

### 4.3. Creation of Artificial Images of AMF

The creation of artificial images makes it possible to investigate the interaction of the foam structure with the additively manufactured structure within an AMF in more detail. For example, it is possible to define the level of porosity for both pore types (foam pores and process-related pores), which allows to compare structures that are difficult to manufacture. The behaviour of the features for these structures provides information on how suitable a feature is for characterising AMF and what conclusions can be drawn from a change.

In order to ensure that artificial images of AMF are comparable to a manufactured AMF, the algorithm must emulate additive manufacturing and the in situ foaming behaviour. The position of the extruded tracks in Figure 1 is not arbitrary; instead, it is determined by the tracks of the slicer software. Accordingly, artificial images of AMF are created by the following procedure, which is additionally illustrated in Figure 4:

The image size and the layer thickness LT are specified in pixels. From this, horizontal lines result that determine the centre of the layers.The specified droplet aspect ratio DAR (parameter of the APF process) and the layer thickness LT can be used to calculate the distance of the tracks *T* according to T=DAR·LT [22], which results in the positions of the tracks within the layers.Ellipses are placed at these positions, which are allowed to deviate randomly from these positions by a small value and are slightly deformed in their shape.Foam pores of a specified size and amount are randomly placed within the ellipses. The pore size and the pore amount are allowed to deviate by a small value.

Following this procedure, the algorithm creates one image at a time. Afterwards, the total porosity and the porosity caused by foam pores are determined on the artificial image and compared with the previously user-defined porosities. If the differences between the desired and actual porosities are below a preset internal threshold, the artificial image is saved, and the algorithm terminates. If the differences are too large, a new artificial image is created, and internal parameters are adapted beforehand. These internal parameters are the width and height of the ellipses, the pore size and the pore amount of the foam pores. However, the position of the ellipses is not changed. For example, if the algorithm has to create more porosity caused by the foam pores, and the total porosity is already set correctly, either the pore size or the pore amount will be increased depending on the setting. However, the size of the ellipses is no longer adjusted, as the total porosity is already set correctly. The algorithm described here can be found on GitHub (https://github.com/IAM-WK/ArtificialAMF.git, accessed on 22 August 2023).

For all created artificial images, a width of 1360 pixel and a height of 1024 pixel was used. For the layer thickness, 40 pixel was chosen, and for the droplet aspect ratio, the same value of 1.25 as used for the manufactured specimens was used.

### 4.4. Determination of Image Features on AMF

In principle, exactly one value is obtained when a feature is determined on an image. In order to obtain a quasi-continuous trend of a feature over an image to localise changes in the AMF, image division is needed (see Section 3). Initially, a section of the image (referred to as a sliding image in the following) is defined. Subsequently, this sliding image is rasterised over the entire image with a certain step size. For each new sliding image, a value of a feature can be calculated and thus, a quasi-continuous trend can be determined.

Figure 5 shows the individual steps of the algorithm used in this work. The algorithm was implemented as a Python Script in Fiji [33] and can be found on GitHub (https://github.com/IAM-WK/AMFeatureCalc.git, accessed on 22 August 2023). The width of the entire image and a changeable height is chosen as the size for the sliding image. In this way, it is possible to measure a quasi-continuous trend of a feature over the specimen height and thus along the layers.

The input is the stitched grey-value image described in Section 4.2. This image is binarised with a local threshold algorithm, which is included in Fiji by default. The threshold method *mean* is used, and the radius parameter of 50 pixel is set for the local domain. An optical comparison between differently binarised images shows that the threshold method *mean* reliably binarises the grey-value images examined. The radius parameter should be larger than the largest image characteristic, such as pores, to avoid incorrect binarisation. The binarisation is intended to remove different illumination characteristics induced by the microscope, as these differences in grey values are unrelated to the information to be examined. In the next step, the sliding image is positioned at the bottom of the stitched image, and for this sliding image, each feature is determined. A loop is now used to move the sliding image up one pixel at a time and determine each feature on the resulting new sliding image. Once the sliding image has reached the top, a quasi-continuous trend is present for each feature over the specimen height. The determination of the features is explained in the following sections.

In this algorithm, the sliding image height (SIH) is a parameter that can be varied. In principle, if the SIH is selected to be too large, different AMFs will be averaged into one value. If the SIH is too small, details in the image and texture information are no longer representative. The SIH must therefore be adapted to the AMF, and this is investigated for the feature pore size, pore amount and entropy by Haralick in this work.

#### 4.4.1. Determination of the Total Porosity

As a result of the binarisation, pores have black pixels and the material has white pixels. Thus, for a 2D image, the total porosity can be defined as the ratio of all black pixels to the sum of all pixels within the sliding image. This ratio is determined for each sliding image.

#### 4.4.2. Determination of Pore Size and Pore Amount

To determine the pore size and the pore amount, all pores have to be labelled. This was realised with the Fiji plugin MorphoLibJ [35]. In some cases, the wall between two individual pores is no longer present because of the abrasion of the thin walls during the grinding process or because they are not visible due to the resolution limitation of the microscope. This makes the segmentation and, thus, the labelling of the individual pores more difficult. Accordingly, two procedures are investigated for the labelling process: in the first procedure, the entire pore area in the stitched image is eroded after binarisation so that connections between the pores are lost. The labelling of the pores is carried out afterwards, and subsequently the pore areas are dilated again so that the feature pore size is not substantially distorted. In the second procedure, this dilation and erosion is omitted. The influence of dilation and erosion on the quasi-continuous trend of the two features is investigated in this work.

After the labelling process, the list of all pores is filtered by pore size. If a pore is smaller than five pixels, it is considered to be part of noise and deleted from the list. For each sliding image, all other pores within a sliding image are used to determine a mean value of the pore size and pore amount. No distinction is made between foam pores and process-related pores, as a visual segmentation is not possible.

#### 4.4.3. Determination of the Entropy by Haralick

In their work, Haralick et al. describe several textural features [29] that are in principle all suitable for the characterisation of AMFs. Previous investigations showed that there is only a small difference between the features recommended by Haralick et al., which is why entropy by Haralick was chosen as an example of a texture feature. The value of the entropy by Haralick cannot be assigned to a physical quantity. If the image has only one grey value, the maximum entropy by Haralick is zero. For images with more grey values, the maximum value increases.

Before the entropy by Haralick can be obtained for an image, the GLCM must be determined (see Section 3). For a binarised black and white image, the GLCM encodes the neighbouring relationship of the pixels in a 2×2 matrix. There are four cases of neighbouring relationships: black–black, white–white, black–white and white–black. The occurrence of these cases is counted on an image and entered into the matrix. This requires the definition of an evaluation direction and a distance to the neighbouring pixel. Then, for each pixel of an image, a neighbouring pixel is searched for in the evaluation direction. For example, for a horizontal direction, the pixels in a row are set in a neighbouring relationship, and for a distance value of one pixel, the neighbouring pixels touch each other. These two parameters, direction and distance, are also investigated for the entropy by Haralick in this work.

For each sliding image, a GLCM is determined, and on this GLCM, the entropy by Haralick can be calculated according to the procedure described in the work of Haralick et al. [29]. Due to the previous normalisation of the GLCM, the entropy by Haralick is independent of the image resolution [36]. This procedure is also implemented in Fiji as a plugin.

## 5. Results

### 5.1. Change of the Features for the Two Manufactured AMFs

In order to compare changes of a feature qualitatively with the AMF, the binarised microstructure is shown in Figure 6 for a masterbatch content of 4% by weight and in Figure 7 for a masterbatch content of 6% by weight next to the feature curves over the specimen height. The value of a feature always refers to the centre of the sliding image, and thus, no value of a feature can be given at the borders of the specimen.

Essentially, the AMF at a masterbatch content of 4% by weight changes at different positions, and different structures exist, while at a masterbatch content of 6% by weight, an optical homogeneous AMF is present. Changes in the structure over the specimen height also represent process changes over the process time due to the additive build-up process. For the AMF at a masterbatch content of 4% by weight, the following changes are visible: In Region 1 between Z = 0 mm to 26 mm, there are many small foam pores and large process-related pores. While the entropy by Haralick and the pore size are almost constant, the total porosity and the pore amount are subject to various changes. This region is similar to the complete AMF at a masterbatch content of 6% by weight. In Region 2 around Z = 30 mm a coarser AMF is visible, and a change in the feature pore size, pore amount and entropy by Haralick is present, while the total porosity remains constant. In Region 3 above Z = 36 mm, changes in the AMF are clearly visible, and all features also change significantly. For the AMF at a masterbatch content of 6% by weight, the features are almost constant.

### 5.2. Influence of SIH, Distance and Direction on the Entropy by Haralick

The parameters SIH, distance and direction influence the behaviour of the entropy by Haralick over the specimen height in various manners. This is demonstrated in the following by using the specimen at a masterbatch content of 4% by weight. As can be seen in Figure 6, the entropy by Haralick is not constant over the specimen height, which makes it more appropriate for investigating parameter influences. Figure 8a shows the entropy by Haralick with varying SIH. SIH is given in layers, and in total, the specimen with a height of 57 mm consists of about 220 layers. Even if there are differences in the entropy by Haralick curve between the three SIH, the minima and maxima are always at the same position. With increasing SIH, the curve becomes smoother and individual minima and maxima disappear.

Figure 8b shows the influence of different distance parameters on the entropy by Haralick curve. As the value for the distance parameter increases, the curve is simultaneously smoothed, and the entropy by Haralick is shifted to higher values. However, the differences in the curve thus become steadily smaller as the distance parameter increases.

The difference in the direction of evaluation is shown in Figure 8c. It can be observed that the entropy by Haralick curves for the vertical and diagonal directions are equal but at different levels. In comparison, the level is more similar for the vertical and horizontal directions, but the entropy by Haralick varies noticeably in Region 1 (see Figure 6) for the vertical direction, while this cannot be observed for the horizontal direction.

### 5.3. Influence of SIH and Dilation and Erosion on the Pore Size and Pore Amount

Also, the parameter SIH and the use of dilation and erosion influence the behaviour of the pore size and pore amount curves over the specimen height. As in Section 5.2, the specimen at a masterbatch content of 4% by weight is used for the investigation of parameter influences.

Figure 9a shows the pore size and pore amount with varying SIH and here, SIH is also given in layers. Similar to the behaviour of the entropy by Haralick with varying SIH, the minima and maxima for the pore size and pore amount are also at the same position, and an increasing SIH smooths the curve over the specimen height.

In Figure 9b, the pore size and the pore amount curves are shown for the case that dilation and erosion are used or not. Regarding the pore size, the use of dilation and erosion smooths the curve and shifts it to smaller values. For both the pore size and pore amount, it applies. Only by using dilation and erosion is the local maximum clearly visible in Region 2 (see Figure 6), and in Region 1, the slope of the two parameters is more constant.

### 5.4. The Behaviour of the Features for Specific Artificial Images

Technically, artificial images are dimensionless. However, they were evaluated in such a way that the layer thickness in pixels corresponds to the layer thickness of 0.26 mm. Using this resulting fictitious resolution, a pore size and pore amount can be given in the same unit as for the manufactured specimens. Since there are no microstructure changes in the artificial images, the features were calculated on the entire artificial image. Dilation and erosion were used to determine the pore size and pore amount, and direction = horizontal and distance = 1 pixel were used for the determination of the entropy by Haralick.

Figure 10 shows three different artificial images of an AMF with different pore sizes and pore amounts of the foam pores. They are not based on any manufactured AMF. In all three artificial images, the total porosity, the porosity caused by the foam pores, and the porosity caused by the process-related pores are kept constant. Only the fineness of the foam structure increases in Figure 10a–c. Table 1 presents the features’ pore size, pore amount and entropy by Haralick for the three artificial images. As desired, the pore size decreases, while the pore amount increases with increasing fineness. The entropy by Haralick increases with increasing fineness.

Compared to the investigation of the fineness of the foam structure in Figure 10, the influence of foaming a track in an additive manufacturing process is investigated in Figure 11. For this purpose, the total porosity is kept constant, and only the porosity caused by the foam pores and the process-related pores is varied.

The three artificial images shown represent a type of AMF that is inspired by the AMF from the Region 3 from Figure 6. Ten different artificial images were created for each type of AMF so that the scattering of the three features entropy by Haralick, pore size and pore amount could be analysed in more detail. Starting from Type 1, the porosity caused by the foam pores is increased by 10% for Type 2 and Type 3, while the porosity caused by the process-related pores is decreased by 10% (total porosity is kept constant). To achieve this increase, the pore amount of the foam pores is increased in Type 2, while their pore size is kept constant. For Type 3, the pore amount is kept constant, and the pore size is increased. It is noticeable that the entropy by Haralick increases with increasing the pore amount and decreases with decreasing the pore amount. Both features behave in a similar way. Starting from Type 1, the porosity in Type 2 can be considered more finely distributed and that in Type 3, more coarsely distributed.

## 6. Discussion

### 6.1. Influence of the Additively Manufactured Structure on the Characterisation of AMF

If Region 1 in Figure 6 is considered in more detail, a repeating pattern in the pore arrangement is noticeable. From Z = 0 mm to 7 mm and between Z = 17 mm to 26 mm, large pores are arranged in vertical strips, while between Z = 7 mm to 17 mm a kind of arrow pattern occurs. The change in the arrangement of the pores can also be seen by the features pore size, pore amount and entropy by Haralick, which periodically show larger and smaller values. This is particularly noticeable for the entropy by Haralick in Figure 8c for the vertical and diagonal directions. In the weak form, a pattern is also visible over the entire AMF in Figure 7. These patterns can be explained by the influence of the additively manufactured structure on the AMF.

Owing to the principle of manufacturing in the MEX and AKF process, process-related pores of different size and shape are formed between the tracks [5]. The size is mainly influenced by the degree of filling [22], while the shape is influenced by the degree of filling and the orientation of the tracks. If a structure as shown in Figure 12 is cut 90° to the orientation of the tracks, the pores ideally appear diamond shaped. In contrast, if cutting is performed along the orientation of the tracks, rectangular pores are obtained between the individual layers.

In plastic-based additive manufacturing, it is common to arrange the tracks in a ±45° orientation to reduce the anisotropy of the mechanical properties [1,3]. Once again, different pore shapes result, depending on the position and orientation of the cutting plane.

This is shown in Figure 13 for two perpendicular cutting planes and one tilted cutting plane. Due to a ±45° orientation of the tracks, the tracks are either cut at intersection points (resulting in pore arrangements in vertical stripes) or outside these points (resulting in an arrow pattern). With a tilted cutting plane, the arrangement of the pores changes periodically from one pattern to the other. The overall pattern shown with the 5°-tilting of the cutting plane in Figure 13c is similar to that of the AMF in Region 1 in Figure 6. It can be concluded that the cutting plane during the specimen preparation is deviated by a few degrees from the desired perpendicular plane of Figure 3a. This demonstrates that the orientation of the tracks in combination with the orientation of the cutting plane can influence the appearance of the process-related pores and thus also the features pore size, pore amount and entropy by Haralick, without having to change anything in the foam structure. Therefore, when characterising AMF, special attention should be focused on an appropriate orientation of the cutting plane so that the shape of the process-related pores does not change as much as possible.

### 6.2. Characterisation of the Fineness of the Foam Structure

In addition to the aspect of the amount by which a plastic can be foamed and what porosity is finally achieved, the knowledge about the fineness of a foam structure is at least as relevant (see Section 2). For example, nanocellular foams are often said to have significantly improved mechanical and functional properties [37]. Thus, the characterisation of the fineness is interesting for AMF as well. From Figure 10a–c, the fineness of the foam structure increases more and more. As expected, for the entire AMF, the pore size decreases, and the pore amount increases (cf. Table 1). This indicates that the two features are still suitable for characterising the fineness of an AMF. The entropy by Haralick can be understood in the presented setting as a kind of weighting from the edges to inner areas. For a binarised black and white image, the size of the GLCM is 2×2. On the track of the matrix, the amount of pixel pairs with the same grey value is encoded: in this case, black–black for pixel pairs within a pore and white–white for pixel pairs within the material. On the diagonal, the amount of pixel pairs with different grey values is encoded. For a parameter distance of one, edges are counted on the diagonal. An edge is defined as the boundary between the material and pore, and in a binarised image, black and white pixels are adjacent to each other. For a foam structure with only a few large pores, the values on the trace are larger than on the diagonal, while for a foam structure with many small pores, the values on the trace and diagonal converge. As the entropy by Haralick for a GLCM with equal entries reaches its maximum value [29], the entropy by Haralick increases with a more finely distributed porosity. Therefore, it can also be observed for the AMFs in Figure 10 that the entropy by Haralick increases with increasing fineness (cf. Table 1).

Thereby, the entropy by Haralick has an advantage over the other two features of pore size and pore amount. While only neighbouring relationships between pixels are taken into account when determining the entropy by Haralick, each pore must be labelled for the determination of the other two features. This procedure sometimes leads to undesirable effects. In Figure 11, the behaviour of foaming in additive manufacturing is emulated. It can be observed that despite the coarsening of the AMF from Type 1 to Type 3, the pore amount increases slightly, and the pore size decreases slightly. An increase in the pore amount and a decrease in pore size with simultaneous coarsening is rather counterintuitive. Using the labelled artificial images for the three different types of AMF in Figure 14, the reason for this can be understood.

In Type 1, a handful of networks of pores occupy almost all of the pore size. The 375 smallest pores contain only about 20% of the pore size, while the remaining 20 pores make up the rest. In Type 3, the pore amount of the foam pores was set to be the same, and their size was increased such that a higher degree of foaming for the tracks could be achieved. This can also be seen from the cumulative pore size—after the 200 smallest pores, the pores of Type 3 clearly increase in size compared to Type 1. In order to realise the increased degree of foaming and a constant total porosity, the porosity caused by the process-related pores was reduced by expanding the tracks in size (for explanation see Section 6.3). This separates the large networks of pores into many smaller networks, which increases the pore amount for the entire AMF. Since the total porosity remains the same, the mean pore size decreases with more pores. In contrast, the refinement of the AMF from Type 1 to Type 2 is reproduced as expected. The pore amount increases, and the pore size decreases. Also from Type 1 to Type 2, the large networks of pores are divided, and the additional increase in the amount of foam pores further increases the pore amount of the AMF. As a result, with more pores and the same total porosity, the pore size is reduced.

It can be concluded that the coarsening and refinement of an AMF can be better represented by the pore size and the pore amount, the less large networks of pores are formed by the process-related pores during the labelling process. In comparison, the entropy by Haralick represents the refinement and coarsening of the AMFs in Figure 11 as expected. However, what entropy by Haralick cannot do is quantify how the shape of the pores changes.

### 6.3. Interaction between the Additively Manufactured Structure and the Foam Structure

In the previous section, the characterisation of the fineness of an AMF is discussed. In addition to this characteristic, another characteristic of a foam structure is the degree of foaming. For some physical properties, such as thermal insulation, the degree of foaming is of particular interest [37]. In principle, the following distinction can be made between the two characteristics (see also Section 2):The degree of foaming is a measure of the increase in volume of an initially non-foamed plastic mass. For foam structures, the porosity can be used to determine the degree of foaming.The fineness of a foam structure indicates how many pores the porosity is distributed over. The porosity remains constant—accordingly, so does the volume of a structure. For example, the pore amount or the pore size can be used as a measure (see Section 6.2).

Both characteristics can change independently but also simultaneously. In a process such as extrusion, the porosity of an extruded strand/profile can be measured to determine the degree of foaming, and the fineness can be evaluated by a microstructure investigation. In Region 3 in Figure 6, a noticeable change in the total porosity can be observed, which leads to the assumption that the degree of foaming has changed. However, a theoretical consideration of the in situ foaming in additive manufacturing shows that a change in the total porosity within the same structure is not plausible. Already in the slicing process, the outer contour of the part is determined by the perimeter. Accordingly, the total volume of the part cannot change any more, unless an overfilling occurs, which is never desirable [22]. Since the mass flow is calculated and fixed in the slicing process independently of the degree of foaming of individual tracks, the mass flow can also be considered constant. Thus, a change in the degree of foaming of the tracks only leads to a change between the porosity caused by process-related pores and by foam pores. The total porosity always remains constant. Figure 2 illustrates this situation. This particularity of AMFs can be proved by the global porosity of the AMF from Figure 6 and Figure 7 (different masterbatch content). Global porosity means that the porosity is not determined on the cutting plane (2D) but on the basis of the entire specimen according to the principle of Archimedes. The global porosity of the AMF from Figure 6 is 24.9% and that of the AMF from Figure 7 is 23.2%, meaning that the two values are quite similar but different from those measured by image-processing methods. To verify this, the global porosities of AMFs with masterbatch contents of 2% (porosity is 24.6%) and 8% (porosity is 24.3%) were also measured. The same process parameters were used for these two new specimens. All values were calculated by determining the bulk density of the mixture of ABS and low-density polyethylene (carrier polymer in the masterbatch) for each masterbatch content. The fact that the total porosity of the AMFs from Figure 6 and Figure 7 differs significantly from each other and that the total porosity changes significantly in Region 3 from Figure 6 can be explained without a change in the degree of foaming of the tracks. This difference could be attributed to the preparation process of the cutting plane, the resolution limit of the microscope and the binarisation.

In the preparation process, the cutting plane must be flattened over a series of grinding steps in order to achieve sufficient image contrast. In this process, edges can be rounded off by grinding particles. Since there are many other edges at the pores, these are also rounded off. This rounding off can be traced in Figure 15 by the darkening grey values starting from the edges.

Furthermore, thin walls between individual pores can be completely removed. After the following binarisation, the total porosity is therefore overestimated due to the unclear focus plane. The resolution limit in turn leads to thin walls being hidden by the surrounding dark grey values. The comparison of the total porosity in Figure 16 for two different resolutions shows that the total porosity is lower for a higher resolution.

It is comprehensible that the accuracy in porosity determination decreases with increasing number of edges, and accordingly the change in total porosity from position Z = 43 mm (porosity about 45%) to Z = 48 mm (porosity about 55%) from Figure 6 can be explained. Although a microscope image cannot contain the information in depth, which is why a difference from a measurement according to the principle of Archimedes is to be expected, the large difference between the two methods can mostly be attributed to the above-mentioned reasons. In summary, the occurrence of the different and changing total porosity in AMFs can be explained despite the equal slicing parameters.

Accordingly, it is impossible to use total porosity to make conclusions about the degree of foaming of individual tracks. If the tracks could be separated within the AMF, it would be possible. However, on the basis of the AMF from Figure 6 and Figure 7, it is comprehensible that this is rarely possible. This also makes it difficult to distinguish between a change in the degree of foaming or only a change in the fineness of the structure since a coarsening of the AMF can also be accompanied by a reduction in the degree of foaming. Nevertheless, there are indicators for when the degree of foaming changes and when it does not. This can be shown by comparing Regions 2 and 3 from Figure 6 with the artificial images from Figure 11. In Region 2, a coarsening of the AMF is observed (e.g., entropy by Haralick decreases), which is comparable to the change from Type 2 to Type 3 in Figure 11. Although a change in the degree of foaming cannot be excluded, it is less probable compared to Region 3. In Region 3, a large increase in the pore size takes place, which is accompanied by a network formation of pores. As can be seen in Figure 11, network formation occurs with a reduction in the size of the tracks, which changes the porosity proportions caused by process-related pores and foam pores. Thus, a change in the size of the tracks is also probable in Region 3, whereby not only the fineness of the structure but also the degree of foaming of the tracks is changed. A possible change in the degree of foaming of the tracks can be explained by the APF process and masterbatch content. Even if the plastic and masterbatch granules are homogeneously mixed beforehand, local differences in the content of the masterbatch can occur due to the small screw flights and the relatively large granules. Since the screw does not have any mixing elements, it is possible to statistically dose plastic quantities that have few to no masterbatch granules. If there are relevant differences in the masterbatch content after the non-return valve, the parts produced in this way will differ in the AMF depending on the current time of the build process. To avoid this, the grain size of the two granules could be reduced, or the suggestion of Nofar et al. to modify the APF process for direct gas injection could be implemented [7].

### 6.4. An Optimal Determination of the Features for Characterisation of AMF

All investigated parameters influence the curve over the specimen height as well as the range of values of the features. How the parameters should be set with regard to a meaningful characterisation of AMF is discussed in the following. 



**Sliding image height (SIH)**



The SIH affects all features in the same way and is relevant for a quasi-continuous evaluation of a feature. Thereby, SIH cannot become arbitrarily small since each individual sliding image must still contain a representative part of information from the entire image. However, if the sliding image is too large, different information will be averaged together. In Figure 8a and Figure 9a, it can be noticed that with increasing the SIH, higher frequency components are attenuated, which is why an increasing SIH acts like a moving average. Specifying the SIH in pixels is not useful due to different image resolutions. For additively manufactured structures, it is practical to specify the SIH in layers. The theoretically smallest possible SIH can be one layer since this way, all different information (tracks, process-related pores and foam pores) is contained in one sliding image. As can be seen in Figure 8a and Figure 9a, the evaluation of changes in the AMF with a SIH of three layers is complicated by the high-frequency components. One of the high frequencies is correlated with the layer thickness and can be given as $f_{LT}=1/0.26$ mm. The effect of abrupt value changes on a feature becomes greater for a SIH with only a few layers, depending on whether the exact number of layers is included in the sliding image or the sliding image intersects two layers. It, therefore, seems appropriate to consider more layers in the evaluation. For the introduced AMFs, a SIH of twelve layers attenuates the frequency $f_{LT}$ sufficiently, while lower frequencies are mostly preserved. If the SIH is increased from 12 to 21 layers, the damping behaviour for the frequency $f_{LT}$ increases only slightly, while the lower frequencies are increasingly damped. This leads to information loss, which can be seen in Figure 9a, for instance, in Region 2 (cf. Figure 6) in the pore size and pore amount. An SIH of 12 layers was found to be suitable for the investigated AMF. However, a characterisation of AMF should be preceded by a study of SIH in order to find an appropriate SIH for the individual problem. 



**Direction and Distance**



For the determination of the entropy by Haralick and the GLCM, two parameters are relevant: direction and distance. As already shown in Section 6.1, the orientation of the cutting plane affects the shape of the process-related pores. Assuming that, despite careful preparation, the effect of changing pore shapes (cf. Figure 13c) cannot be completely avoided, the horizontal direction, and therefore parallel to the layers, has an advantage for the characterisation of AMF. This way, there is no effect on the formation of neighbouring pixels whether process-related pores are shaped as in Figure 13a,b. In comparison, it can be seen from Figure 8c that the effect on the entropy by Haralick is considerable with a direction that is vertical or diagonal. Therefore, the horizontal direction is recommended. The parameter distance specifies the distance at which neighbouring relationships are formed between two pixels. If the parameter is set to one, then adjacent pixels are placed in a neighbouring relationship, which can be seen as a kind of edge detector. If the parameter distance is increased, then pixels are set in a neighbouring relationship with each other, which actually does not correspond to any edge. Considering Figure 8b, it can be observed that with increasing the parameter distance, the entropy by Haralick is shifted to higher values, and the differences between different AMFs become smaller. Consequently, the values of the trace and diagonal of the GLCM become more similar (see explanation in Section 6.2). Clearly, a larger parameter distance leads to a more random clash of black and white pixels. Such random behaviour is not desired, which is why the behaviour of an edge detector and thus a distance parameter of one is recommended. 



**Dilation and Erosion**



For the labelling process and thus for the determination of the pore size and pore amount, the use of dilation and erosion is essential. Without this processing step, a small connection between pores leads to large networks of connected pores. If such a large network lies in the sliding image or leaves it, this leads to large changes in the pore size. This can be observed, for example, between Z = 14 mm to 19 mm in Figure 9b.

Figure 17 shows the AMF from this section as well as the labelled image with and without the use of dilation and erosion. Without dilation and erosion, large networks of process-related pores can be seen. Such large changes in the value of the two features hide real structural changes. Hence, the structural change in Region 2 (cf. Figure 6) is not identifiable without the help of dilation and erosion. Accordingly, this processing step is necessary for a correct characterisation of AMF, although not all connections between pores can be avoided as shown in Section 6.2.

## 7. Conclusions

Additively manufactured foam structures are a relatively new field in plastic-based additive manufacturing. Accordingly, there are only a few works in this field, and no one deals with the optical microstructure characterisation of such structures. In this work, typical features (total porosity, pore size and pore amount) as well as a texture feature (entropy by Haralick) were determined to further investigate the interaction of the foam structure with the additively manufactured structure. For example, this will enable future research to characterise additively manufactured foam structures in terms of parameter influences.

For this purpose, specimens of acrylonitrile–butadiene–styrene and a masterbatch (containing a chemical blowing agent) were manufactured by using the Arburg plastic freeforming process, and microscope images were taken of their inner structure. Furthermore, artificial images of additively manufactured foam structures were made to enable the characterisation of structures that are difficult to manufacture. The features were determined on these manufactured and artificial images, whereby a quasi-continuous trend was determined for larger structures.

It was shown that the additively manufactured structure alone can cause different pore sizes and a different pore amount due to its process-related pores in combination with the orientation of the cutting plane. Depending on the orientation of the cutting plane, this effect occurs with a different periodicity. Therefore, caution is required when identifying changes within the foam structure. Furthermore, it could be shown that with equal slicing parameters, the total porosity is always the same—regardless of whether the foaming process has changed temporarily. The reason for this is the way the plastic-based additive manufacturing takes place. The mass flow and the external volume of the part always remain constant, and only the ratio between process-related pores and foam pores changes when the foaming process is changed. For structures with a low amount of process-related pores, it is difficult to identify individual tracks. As a result, a change in the degree of foaming of an additively manufactured foam structure can no longer be distinguished from a change in the fineness of the foam structure. However, on the basis of the artificial images, an indicator could be found that implies a change in the degree of foaming. The formation of large networks of process-related pores indicates a decrease in the degree of foaming of the tracks. In contrast, the fineness of a foam structure can be well characterised, although it should be ensured that no large networks of process-related pores occur. Finally, suggestions for an optimal determination of each feature investigated were given. The conclusion for each feature is as follows:The feature total porosity is only suitable for the validation of whether the porosity specified in the slicing process has been achieved. It cannot be used to observe changes in the foaming behaviour of an additively manufactured foam structure.The features pore size and pore amount can be used to characterise the fineness of a foam structure. Furthermore, they can be used to identify networks of process-related pores. These are an indicator for a change in the degree of foaming.The feature entropy by Haralick is generally well suited for identifying changes within additively manufactured foam structures, and wrong estimations can be excluded. It can therefore be used to first identify factual changes and then investigate them with further features.

## Figures and Tables

**Figure 1 polymers-15-03544-f001:**
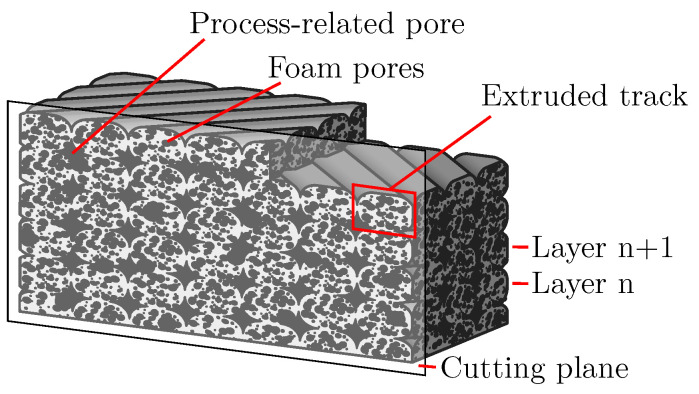
Sketch of an additively manufactured foam structure (AMF). The tracks are orientated layer by layer in a ±45° orientation, resulting in diamond-shaped process-related pores in this cutting plane. Foam pores are present within the tracks.

**Figure 2 polymers-15-03544-f002:**
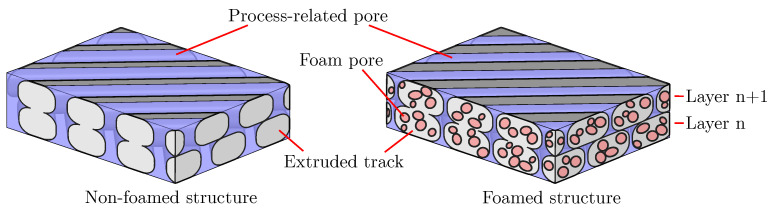
Sketch of two additively manufactured structures. The position of the tracks is the same in both structures, whereby the tracks of the right structure are larger and contain foam pores due to the foaming process. The tracks are orientated layer by layer in a ±45° orientation.

**Figure 3 polymers-15-03544-f003:**
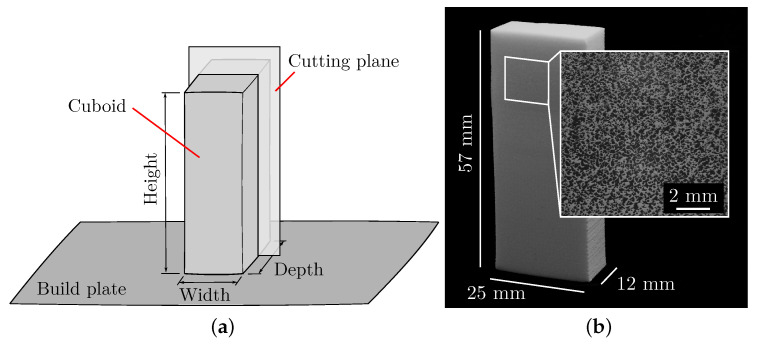
Preparation of the specimen with an AMF. (**a**) Sketch of the cuboid on the build plate and the position and orientation of the cutting plane. (**b**) Cut cuboid with ground and polished surface. A section of the stitched microscope image is shown to the right.

**Figure 4 polymers-15-03544-f004:**
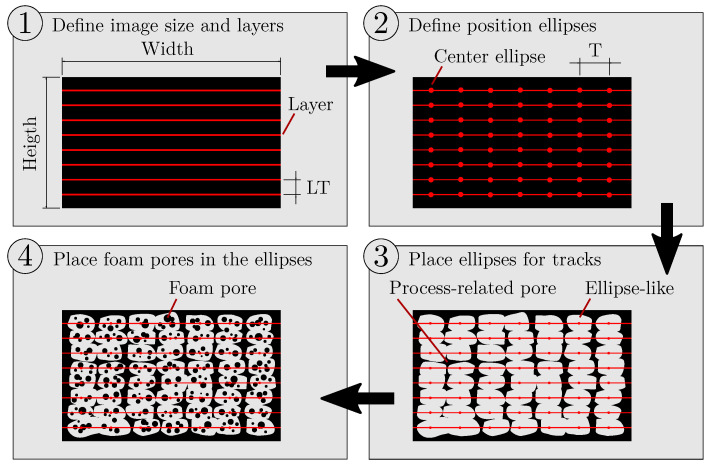
Four-step procedure to create artificial images of AMF. Layers are placed at the distance of the layer thickness LT, and deformed ellipses are placed within the layers at the distance of tracks *T*. Within the ellipses, foam pores are placed with variations in pore size and pore amount.

**Figure 5 polymers-15-03544-f005:**
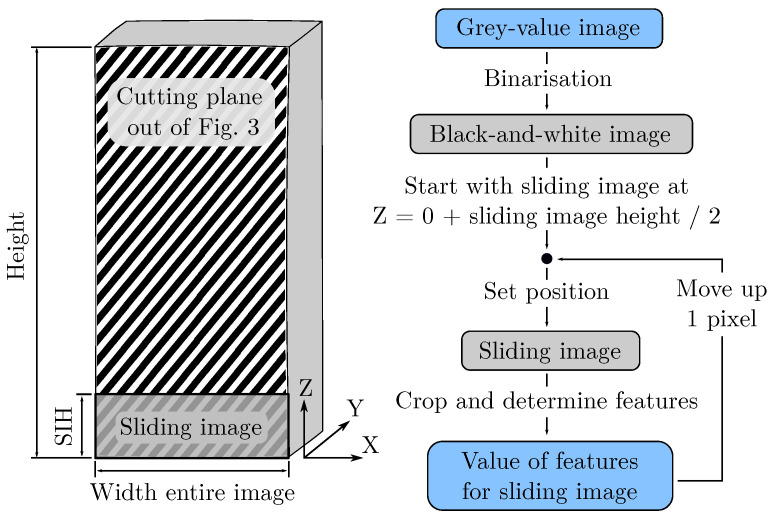
The AMF is characterised on the shaded surface of the specimen. The sliding image is a section of the entire image, which is moved up along the Z-axis. The procedure of the algorithm is shown to the right. For each sliding image, a value of a feature is determined.

**Figure 6 polymers-15-03544-f006:**
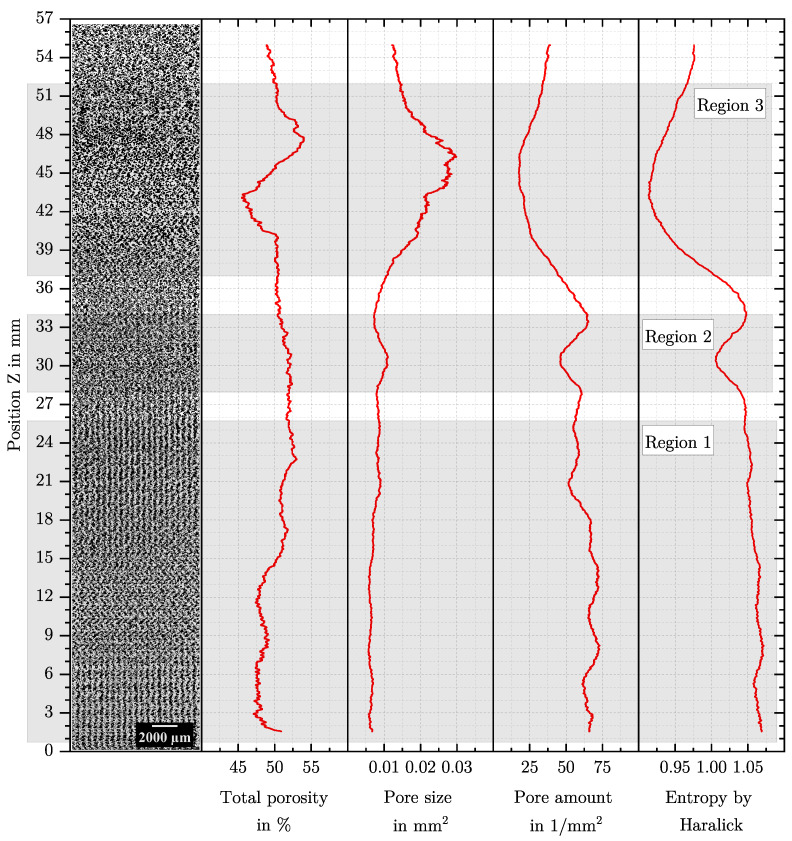
Comparison between the four investigated features total porosity, pore size, pore amount and entropy by Haralick for the AMF at a masterbatch content of 4% by weight. The binarised microstructure is shown next to the features. Qualitatively different AMFs can be identified. Three noticeable regions are marked with a grey box. SIH = 12 Layer was used for all features. Dilation and erosion were used to determine pore size and pore amount, and direction = horizontal and distance = 1 pixel were used for the determination of the entropy by Haralick.

**Figure 7 polymers-15-03544-f007:**
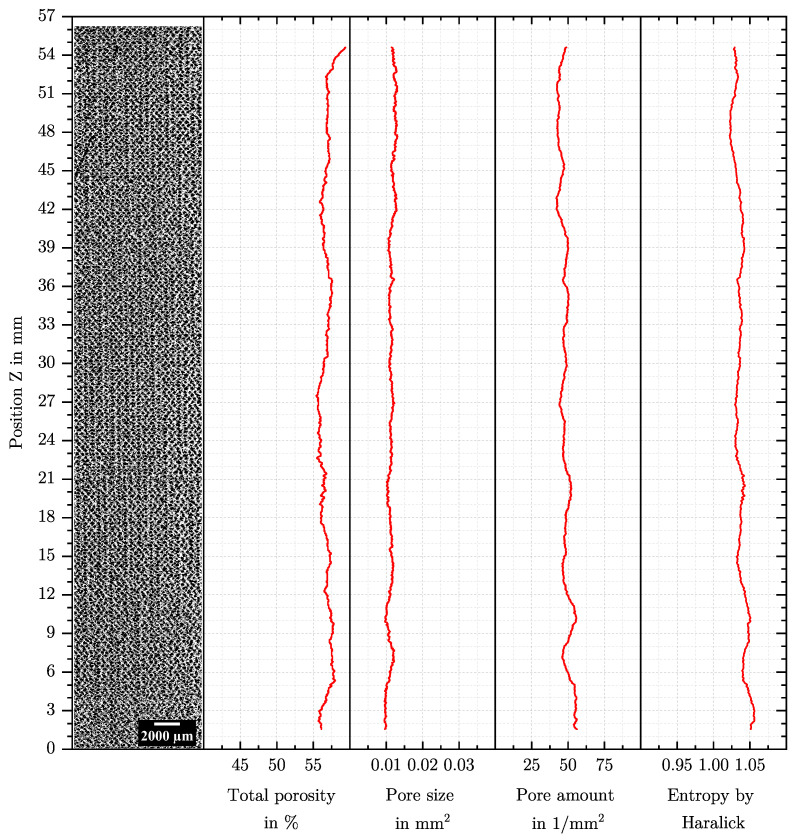
Comparison between the four investigated features total porosity, pore size, pore amount and entropy by Haralick for the AMF at a masterbatch content of 6% by weight. The binarised microstructure is shown next to the features. There is no discernible difference in the AMF. SIH = 12 Layer was used for all features. Dilation and erosion were used to determine pore size and pore amount, and direction = horizontal and distance = 1 pixel were used for determination of the entropy by Haralick.

**Figure 8 polymers-15-03544-f008:**
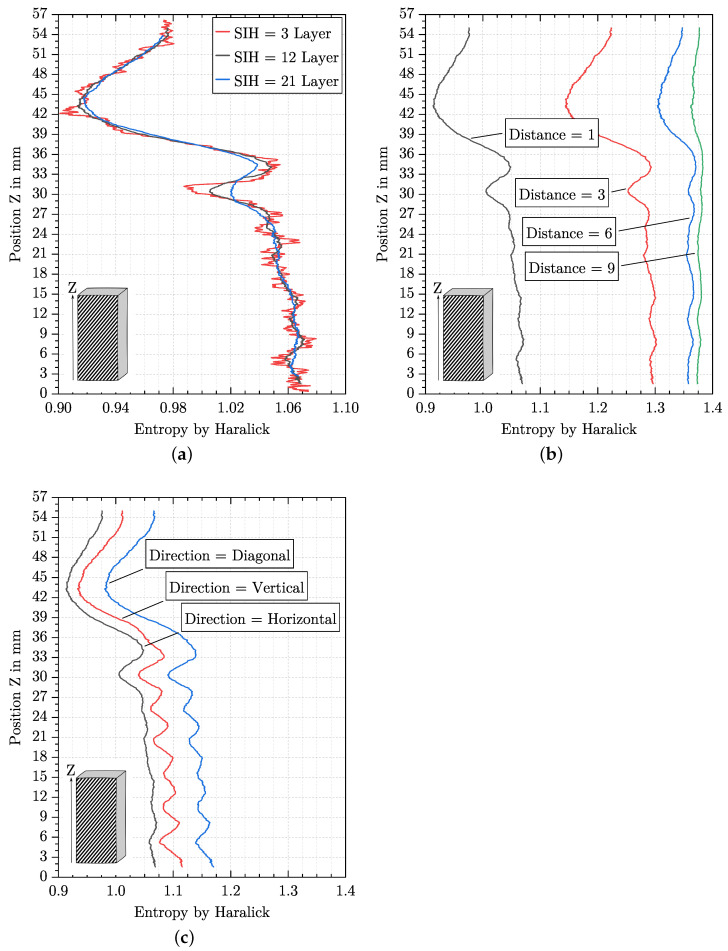
Entropy by Haralick over the specimen height at a masterbatch content of 4% by weight. Offset of curves at the top and bottom due to the value of SIH. (**a**) Three different SIH are shown and their influence on the entropy by Haralick. The parameters direction = horizontal and distance = 1 pixel were used. (**b**) Influence of four different distance parameters on the entropy by Haralick curve. The parameters direction = horizontal and SIH = 12 Layers were used. (**c**) Different entropy by Haralick curves over the specimen height for three different direction parameters. The parameters distance = 1 pixel and SIH = 12 Layers were used.

**Figure 9 polymers-15-03544-f009:**
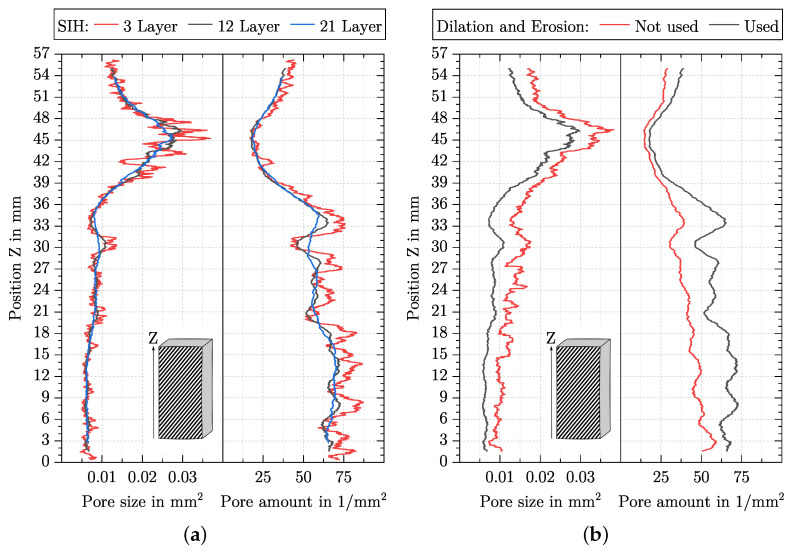
Pore size and pore amount over the specimen height for the specimen at a masterbatch content of 4% by weight. Offset of curves at the top and bottom due to the value of SIH. (**a**) Three different SIH are shown as well as their influence on the pore size and pore amount. Dilation and erosion were used. (**b**) Influence of dilation and erosion on the pore size and pore amount. The parameter SIH = 12 Layers was used.

**Figure 10 polymers-15-03544-f010:**
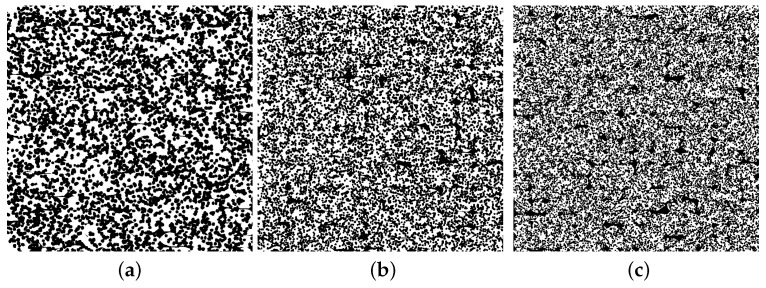
Three artificial images of an AMF. While the total porosity and the porosity caused by the foam pores are constant, the pore size and pore amount of the foam pores vary. The total porosity is about 51%, and that of the foam pores is about 42%. (**a**–**c**) The fineness of the foam structure increases.

**Figure 11 polymers-15-03544-f011:**
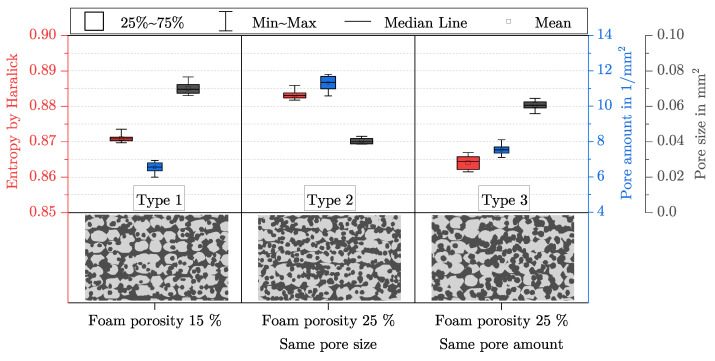
Scattering of entropy by Haralick, pore amount and pore size for three types of AMF. For each type, ten artificial images (one artificial image is shown as an example) were created with a constant total porosity of about 45%. All specifications under the artificial images refer to the foam pores. Starting from Type 1, the porosity caused by the foam pores is increased in Type 2 and Type 3, once the pore amount and the pore size of the foam pores are kept constant.

**Figure 12 polymers-15-03544-f012:**
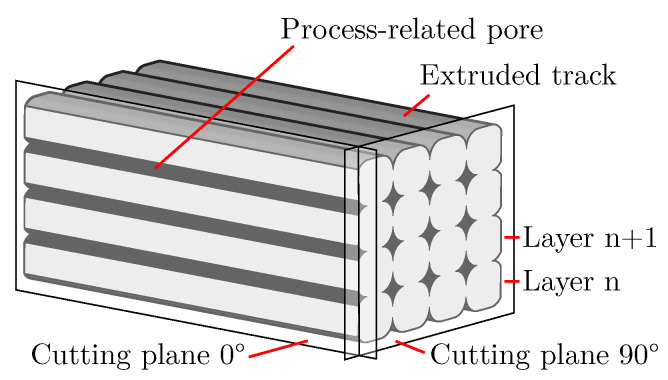
Sketch of an additively manufactured structure. All tracks are orientated in one direction. Due to the orientation of the cutting plane relative to the track orientation, the shape of the process-related pores varies from rectangular to diamond shaped.

**Figure 13 polymers-15-03544-f013:**
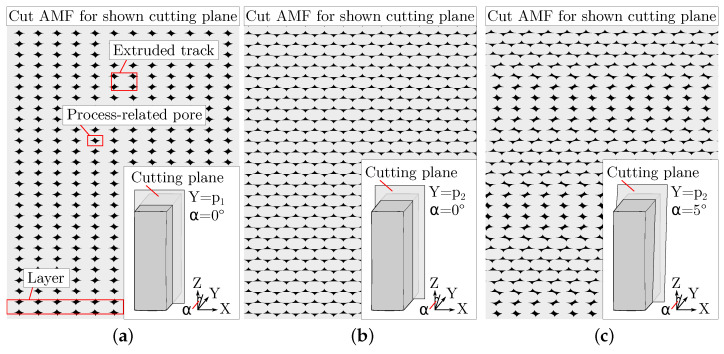
Three cutting planes at different positions and with different orientations on a virtually created model of an additively manufactured structure with a ±45° orientation of the tracks. (**a**,**b**) The resulting arrangement of the process-related pores for a perpendicular cutting plane and (**c**) the resulting arrangement for a tilted cutting plane.

**Figure 14 polymers-15-03544-f014:**
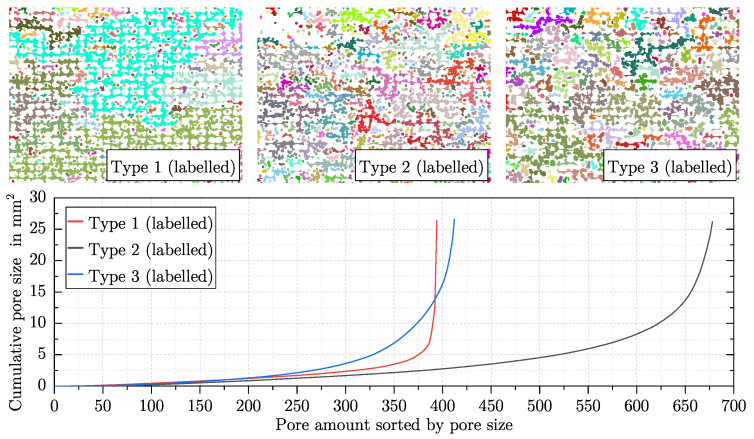
An example of each type of artificial image from Figure 11 is shown, whose pores are labelled. The colour marks the affiliation of a pixel to a pore. More distant pores without a connection can share the same colour. In addition, the pore size is shown cumulatively above the pore amount for each example. The pores are sorted by size so that the contribution of individual pores to the total porosity can be identified.

**Figure 15 polymers-15-03544-f015:**
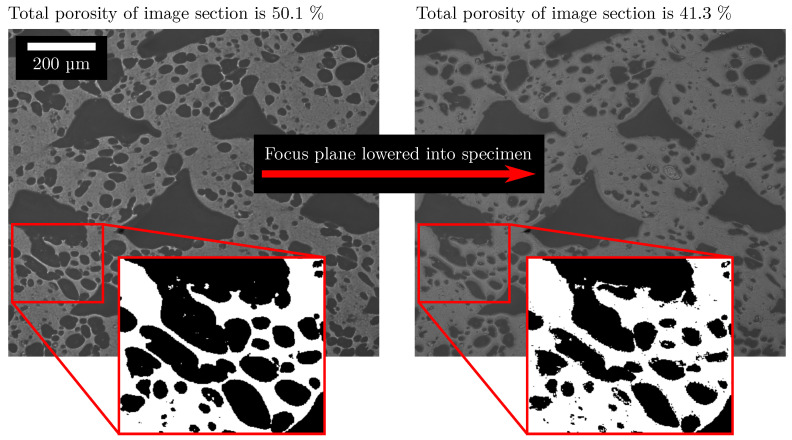
Two microscope images with a higher resolution than those used for determining the features. The focus plane was varied to show the effect of rounding off edges. The influence on the separation between material and pores can be traced on the basis of the binarised section. The total porosity is clearly different. The image section is from the AMF from Figure 7.

**Figure 16 polymers-15-03544-f016:**
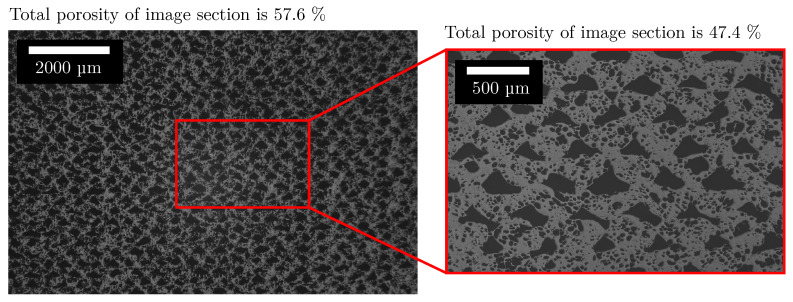
Two microscope images with different resolutions. The resolution of the left microscope image corresponds to that used for determining the features. Due to the lower resolution, small walls between pores are no longer represented, which increases the total porosity. The image section is from the AMF from Figure 7.

**Figure 17 polymers-15-03544-f017:**
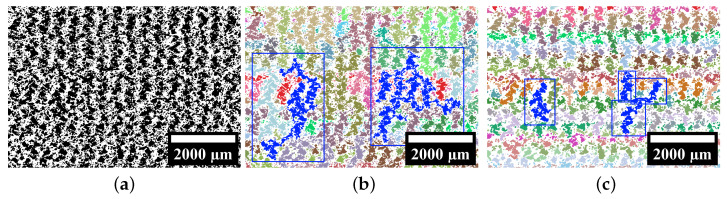
Image section taken from the entire AMF from Figure 6. The section can be found between Z = 14 mm to 19 mm. (**a**) Image section is shown binarised. (**b**) The same image section with labelled pores. The colour marks the affiliation of a pixel to a pore. More distant pores without a connection can share the same colour. Some highly enlarged pores are marked in blue. (**c**) The same image section with labelled pores using dilation and erosion. The enlarged pores are divided.

**Table 1 polymers-15-03544-t001:** Values of the features for the three artificial images from Figure 10.

Artificial Image of AMF	Entropy by Haralick	Pore Size in mm^2^	Pore Amount in 1/mm^2^
Fineness: Coarse (Figure 10a)	0.955	0.0200	25
Fineness: Intermediate (Figure 10b)	1.057	0.0055	90
Fineness: Fine (Figure 10c)	1.131	0.0023	214

## Data Availability

The data presented in this study are available on request from the corresponding author.

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
