# Peer review of "On the Creation and Optical Microstructure Characterisation of Additively Manufactured Foam Structures (AMF)â€"

_polymers, 2023, doi:10.3390/polym15173544_

Round 1
Reviewer 1 Report
This work addresses a very interesting issue with the characterization of multi-scale porosities in a cellular structure made of porous materials (in this case using material extrusion). The work focuses on the investigation of cross section image-based porosity characterization, and introduced an additional term of Haralick entropy in addition to the existing characteristic sets, in order to capture the heterogeneous nature of the multi-scale porosity with material extrusion-based foams.
That said, the paper is in general difficult to follow, and for a work that aims to provide additional characterization concepts, its explanation and exploration of concepts are somewhat confusing. This should likely be checked and improved upon revision.
Some additional comments:
The term “fused filament fabrication” should be changed to material extrusion, per the ASTM F2797.
In the introductory discussion about the use of AM for the fabrication of porous structures, there exist many specific statements/claims that do not appear robust or correct. One example is the claim that for thin-walled structures, it is infeasible to generate large porosity by increasing inter-track distance. Another example is the discussion about cellular solids and foams as two different categories, while in fact there exist significant overlap in their definitions.
In the introduction, the reason why the proposed study is necessary was not well-clarified. The argument that there exist material extrusion process-induced porosity (e.g. gaps between tracks) and “embedded” porosity from the filament is legitimate, however it is unclear why the two need to be differentiated, as well as the respective challenges and issues for the adjustment of the two types of porosities. In addition, part of the lack of clarity might also come from the use of terms and phrases. For example, the porosity within the filament should not be referred to as “structure”, and as the AM process is the focus of the study, the word “foaming” should be avoided, as it refers to the process of introducing stochastic gas phases in the liquid material via traditional processes.
In the discussion the authors suggested that due to the similar granular sizes of the two materials, segregation will not occur. However, the flow of particles in a powder bed is not only influenced by particle size but also the density of the material and the surface characteristics of the particles (e.g. friction coefficient). Therefore such statement does not appear to be accurate.
For easier visual presentation, it might be more efficient if Fig.5 and Fig.6 can be laid out side-by-side.
In the discussion about the effects of different modeling parameters on the results, the authors presented detailed comparison, but there is still a lack of connection between these discussions and the suitability of the parameter levels. Thus the choice of parameter setting in the later sections is not well-justified.
In the discussion about the effect of embedded porosity (in filament, or “foaming” in this work), it is unclear whether the “type” of porous structures here correspond to the 3 regions from Fig.5, or were all 3 types of samples generated with the parameters from Region 3. The discussion must be improved for clarity.
The treatment of porosity generation parameters for Type 2 and Type 3 artificial samples is confusing. If material porosity level is reduced by 10% for Type 3 (“reversed”), then how could the overall porosity level remain constant? The discussion here likely needs to be rephrased to avoid confusion.
From Fig.10, it does not appear that Type 2 (middle) exhibits the same level of pore size (black bar) as Type 1. Same issue with the pore amount between Type 3 and Type 1. The caption/discussion need to be revised to clarify about this and to avoid confusion.
The discussion about the relationships between the process-induced porosity and the material porosity (Section 5.3) is very confusing and difficult to follow. It is unclear why constant porosity was set as the condition of analysis; it is unclear why the “total volume” of the process cannot change, given that the spacing between the tracks can always be adjusted to reduce the volume of material. If the process parameters remain identical, then wouldn’t the change of filament porosity result in the increase of overall porosity?
The discussion about the reason of different porosity levels at different Z position with the 4% sample (Fig.5) appears to suggest that the resolution of the observation was the primary source of deviation. However this would suggest that the actual porosity level would remain identical. Also, I don’t follow how this discussion is related to the conclusion that the overall porosity level could not be employed to characterize the filament/track porosity level. This particular conclusion should be self-apparent from the definition of the overall structural porosity.
I think that the terms that are employed to define the different types of porosities, such as “fineness”, “degree of foaming”, and “porosity”, must be very clearly defined and used consistently. Although I do not feel that the choice of names is well-conceived for easy understanding, they could still be used in this work as long as they are clearly explained. As it current is, it is challenging for me to follow many of the discussions in this work.
I would suggest extensive check with the writing style. The discussions are difficult to follow.
Reviewer 2 Report
Reviewed paper is well prepared. Introduction part has sufficient information, methods are adequatly described.
Topic of conducted research is connected with modern trend in technology - modelling. Author's experiments concentrate on developing, using, improving and testing of foam structures manufacturing.
Such experiments can be useful and paper can be published after few minor corrections listed below.
1. There should be outline of paper described at the end of first chapter
2. Algorithm and script described in chapter 3 are available on GitHub, did authors developed them or just use to make experiment? Full graph of algorithm should be presented.
3. I didn't find in conclusions information about practical applications. Did authors veryfied proposed method in practice? I.e. did authors try to produce foam structure based on the model? How efective was model, did the structure physical/mechanical parameters were accepted?
Round 2
Reviewer 1 Report
For Fig.10, the label for the x-axis should be “foam porosity”.